# High Affinity Aptamers and Their Specificity for Azaspiracid-2 Using Capture-SELEX

**DOI:** 10.3390/md23050183

**Published:** 2025-04-25

**Authors:** Jiaping Yang, Xinhao Li, Weiqin Sun, Yunyi Cui, Han Chen, Yao Yang, Mingjuan Sun, Lianghua Wang

**Affiliations:** Basic Medical College, Naval Medical University, Shanghai 200433, China; smmu_shenghuayjp@163.com (J.Y.); li95613@163.com (X.L.); qinjiao1225@163.com (W.S.); yunyicui@163.com (Y.C.); nmuchenhan@foxmail.com (H.C.); yangyao2017@foxmail.com (Y.Y.)

**Keywords:** azaspiracids, nucleic acid aptamers, molecular dynamics simulation, biolayer interferometry, biosensors

## Abstract

Azaspiracids are a type of polyether toxin. Currently, the existing detection methods for azaspiracids all have certain drawbacks. Aptamers offer a cost-effective and convenient approach for the detection of azaspiracids. By employing the Capture-SELEX (Systematic evolution of ligands by exponential enrichment) method to screen aptamers specific to azaspiracid-2, a high-affinity aptamer can be identified for toxin detection. The bin ding affinity of the toxin is verified using biolayer interferometry (BLI) technology. Additionally, computer simulations are utilized to explore the binding sites of the aptamer and conduct molecular dynamics simulations to investigate the stability of the aptamer–toxin complex. Further optimization of the obtained aptamers is carried out to enhance their affinity for the toxin. Ultimately, two aptamers, JD2-RM3-27C28T and JD3-RMM1, are obtained, with dissociation constants (*K*_D_) improved by two orders of magnitude (*K*_D_ = 8.7 × 10⁻⁸ M and *K*_D_ = 6.8 × 10⁻⁸ M, respectively). These aptamers have the advantage of being incorporated into a new AZA2 assay that is more accurate and ethical than biological monitoring methods, and more economical than LC-MS. In the future, this is expected to demonstrate significant advantages in the fields of food safety, environmental toxin monitoring, toxin exposure diagnosis, and public health monitoring.

## 1. Introduction

Azaspiracids (AZAs) are a class of lipophilic polyether and diarrheal marine toxins. They are currently known to be generated in the native organisms *Azadinium* [1,2] and *Amphidoma* [3,4] of the flagellate order, and certain species of *Protoperidium*, such as *P. crassipes* [5] (all of which belong to the phylum dinoflagellate), and can bioaccumulate in shellfish, such as oysters, scallops, clams, etc. [6,7,8]. It was first reported in 1996 during a food poisoning incident that occurred in Ireland in 1995, where several crew members developed acute gastrointestinal symptoms after consuming the blue mussels they had captured [9]. The toxin was initially named Killary Toxin (named after its discovery site) or KT-3, and later renamed Azaspiracids to better reflect its structural formula [10]. The physicochemical properties of AZAs toxins are significantly different from other nitrogen-containing biotoxins. As with other marine toxin groups, the toxins are relatively stable and cannot be eliminated by conventional cooking and processing. After several hours of high-temperature treatment with various organic reagents, the structure and toxicity of the azaspiracids remain unchanged, and they can be stored for a long time under refrigeration conditions [11]. At present, more than 60 types of AZAs and their derivatives have been discovered, and most of which are derived from the secondary metabolites of mussels [12,13].

There is currently no definitive experimental study on its pathogenic mechanism that can demonstrate it as a blocking agent for a specific target. However, some studies have shown that it can influence the electrical activity of cells by affecting the activity of various ion channels, including potassium ion channels [14], sodium ion channels [15], chloride ion channels [16], calcium ion channels [17], and so on. For the impact on the individual digestive system, toxins can enter the bloodstream through the human intestinal barrier, causing damage to epithelial cells [18]. It also has certain effects on inducing tumors [19], altering the cytoskeleton, and reducing human cell metabolism [20]. Experiments have shown that repeated exposure to azaspiracids can induce the expression of Jun B proto-oncogene, Jun D proto-oncogene, c-Jun proto-oncogene, FBJ murine osteosarcoma viral oncogene homolog B, and Fos-related antigen 1 in mice, leading to a significant increase in lung tumors and lymphatic necrosis of digestive system tissues [21]. In addition, there have been reports on the impact to the nervous system [22].

The European Food Safety Authority (EFSA) established an acute reference dose (ARfD) at 0.2 μg/kg body weight (b.w.) based on data from human poisoning incidents [23] and recommended a guideline limit of 30 µg/kg maximum concentration in shellfish meat [24]. However, European Law states that the maximum permitted threshold for total azaspiracids is 0.16 mg/kg AZA-1 equivalents [25].

The main detection methods currently include the following aspects:Mouse bioassay (MBA) involves injecting toxins or extracts into the abdominal cavity of mice and recording the survival time of mice at different concentrations to calculate the toxin content of the sample. This method is the simplest and most commonly used method for detecting shellfish toxins, with a detection sensitivity of 1 μg level [26]. The minimum lethal doses for purified AZA-1, AZA-2, and AZA-3 are 200 μg/kg, 110 μg/kg, and 140 μg/kg, respectively [27]. This method has been listed as a routine method for toxin detection by the American Society of Analytical Chemists. The advantage is that it does not require complex equipment and has high operability. However, there are also drawbacks such as poor sensitivity, low reproducibility, high false positive rate, inability to identify toxicity, and significant influence from individual differences in mice.Liquid chromatography tandem mass spectrometry (LC-MS/MS) was first used in 1999 to detect protochloroalginic acid, with a detection sensitivity of up to 50 pg [28]. This method is the European Reference Method for determination of AZAs, with advantages such as high sensitivity, high specificity, and the ability to accurately determine the composition of mixtures. However, there are still drawbacks such as expensive instruments and the need for skilled technicians to operate them.Enzyme-linked immunosorbent assay (ELISA) has a detection sensitivity of 0.45–8.6 ng/mL [29]. Using new plate-coating antigen, an ELISA was developed with a working range of 0.30–4.1 ng/mL and a limit of quantification at 37 μg/kg for AZA-1 in shellfish [30]. ELISA can also be employed in conjunction with gel methods [31]. This method may have adverse effects such as false positives and cross-reactivity. And currently, there is no commercial ELISA available for purchase and use.Biosensors: Current monitoring methods are limited to the detection of standard samples, including polyclonal antibody recognition [32] and monoclonal antibody recognition [33] targeting the C28–C40 site FGHI loop. However, there are still issues with the toxicity and lethality of immune conjugates in vivo. Other biological assays, including magnetic bead based, chromatographic column based, and microsphere immunoassay [34], have provided new detection approaches.

These detection methods each have their own advantages, but most still have many obstacles in application. As a novel molecular recognition tool, nucleic acid aptamers have multiple significant advantages, with clear origins and a wide range of applications. The word comes from the Latin word “aptus” (adaptation) and the Greek word “meros” (partial) [35]. It is mainly composed of DNA and RNA, and aptamers form structured single-stranded nucleic acid secondary structures through hydrogen bonding interactions between bases, such as hair clips, loops, inner loops, multi-loops, protrusions, stems, pseudoknots, and non-standard base pairing. On the basis of the secondary structure, they can also form certain pocket structures by curling and folding themselves, or form guanine-rich triple- or quadruple-stranded structures through layer stacking [36]. Small molecules can be recognized by nucleic acid aptamers through electrostatic binding, groove surface binding, insertion binding, and other methods [37]. Electrostatic binding mainly occurs when the small molecule ligand is a positive dot molecule, and the binding occurs through charge adsorption. The electrostatic effect is very weak, and it is only meaningful for the complex to bind to the aptamer in the presence of a large distribution of positive charges. The specificity of simple small molecule binding is poor; groove surface binding mainly refers to the groove binding formed between ligands and hydrogen bond surfaces or multiple pairs of nucleic acids; The structure formed by stacking π-bonds and basal planes in compounds with smaller molecular weights using insertion binding design is the most stable [38]. After partial screening and optimization, the equilibrium dissociation constant *K*_D_ value of some aptamers can reach between nM and pM, which makes them highly accurate in molecular recognition [39,40,41]. Nucleic acid aptamers have demonstrated extremely high advantages in low immunogenicity, low production cost, stability after modification, small batch differences, small volume, convenient transport, resistance to pH and temperature-induced denaturation, ease of modification with nanomaterials or organic dye molecules, and flexibility in engineering performance manipulation [42]. For some target nucleic acid aptamers, they exhibit higher affinity and sensitivity than proteins while also being able to recognize hidden binding sites that are inaccessible to smaller targets or protein antibodies [43]. Nucleic acid aptamers can be combined with various signal amplification techniques to achieve high-sensitivity detection of target molecules. One of its advantages is strong adaptability, which can be applied to various monitoring scenarios such as biosensing, environmental monitoring, food safety, clinical diagnosis, and nucleic acid aptamers. After binding to the target molecule, it can be dissociated under denaturing conditions (such as high temperature, pH changes) to restore its activity, achieve reuse, and reduce detection costs. With the development of experimental technology, aptamers are expected to replace enzyme-linked immunosorbent assays as an effective method for the detection of various chemical molecules.

Azaspiracids exhibit significant cytotoxicity and cardiotoxicity, posing a direct threat to consumer health [21]. With global climate change and deterioration in the marine environment, harmful algal bloom (HAB) incidents occur frequently [44], leading to an expansion of the contamination range of AZA-2. The European Food Safety Authority (EFSA) has reported multiple shellfish recall incidents caused by AZA-2 contamination, highlighting the necessity for strengthening detection efforts [45]. Furthermore, AZA-2 is highly stable in the environment and difficult to remove through conventional water treatment processes. Its bioaccumulative properties may result in the gradual amplification of toxin concentrations within marine ecosystems [46]. The establishment of environmental benchmark values and ecological risk assessment models for AZA-2 relies on large-scale and high-frequency monitoring data, which current monitoring capabilities struggle to meet. Therefore, more novel and efficient detection methods are urgently needed.

The experimental plan is to screen for aptamers of Azaspiracid-2 using the Capture-SELEX method. To find one or more high-affinity aptamers for toxin detection, BLI technology is used to verify the affinity of the toxin. Additionally, computer simulation is used to investigate the stability of the aptamer docking site and molecular dynamics simulation to improve the affinity of the aptamer for the toxin. The results ultimately achieve specific detection of Azaspiracid-2 in the sample.

## 2. Results

### 2.1. Capture SELEX Method for Obtaining AZA-2 Aptamers

Capture-SELEX includes the following steps, which require several cycles before sequencing, as shown in Figure 1. Starting from the second round, the library was obtained through a long short-chain separation method. After high-temperature denaturation, two bands appeared on the denatured nucleic acid electrophoresis gel. The band far from the sample site was selected as the band for the next library round (Figure 2A). Successful Capture-SELEX relies on efficient library binding, and the binding rates for each round of library screening were all above 75%, as shown in Figure 2B. If the binding efficiency of the library is lower than expected, methods such as extending the incubation time, adjusting the library capture sequence ratio, and increasing the number of magnetic beads should be used to optimize the reaction conditions.

In the elution step, AZA-2 targets were used as forward screening targets in rounds 1–12, and okadaic acid (OA) targets were used as reverse screening targets in rounds 6–12 to increase screening pressure and improve aptamer specificity. There are also ways to increase screening pressure by reducing the number of magnetic beads used, lowering the initial library concentration, reducing the co-incubation time of library magnetic beads, and prolonging the cleaning time for the magnetic beads after incubation. The recovery rate for each library round during the screening process is shown as an important monitoring indicator in Figure 2C.

High throughput sequencing was performed on the eluate after 12 rounds of toxin mixing, and a total of 1,048,575 differential sequences were displayed. Compared with the theoretical value of the initial library, the number of library readings decreased by about 70%, from 88% to 13% (Figure 3A). The library lengths of the top 100 sequences ranked in the 12th round of high-throughput sequencing are the same as the original length of the initial library, indicating the effectiveness of high-throughput sequencing. The top ten sequences (Table 1) in the entire library increased from 0.78% in the first round to 12.60% in the 12th round (Figure 3B) (JD-1 to JD-10 represent the rankings from 1st to 10th in terms of read counts from sequences obtained after removing primers at both ends in the final round of Capture-SELEX).

Biolayer interference (BLI) technology is a label-free optical biosensing technique used for real-time monitoring and analysis of biomolecule interactions, along with quantitative analysis of their binding strength and dynamics [47]. This technology is based on the principle of light interference, using biosensors to couple nucleic acid aptamers or proteins to probes at the end of the sensor. The probes are immersed in the sample to capture the analyte and includes experimental steps such as sample loading, ligand binding and fixation, target binding, elution, and analysis (Figure 4). Preliminary affinity detection was conducted for the aptamers screened in the early stage, and the basic BLI data for the truncated aptamers are presented in Table A1. Among them, the JD-2 aptamer from the JD-2 family (with a *K*_D_ value of 3.5 × 10^−6^ M) and the JD-3 aptamer from the JD-3 family (with a *K*_D_ value of 3.6 × 10^−6^ M) (Table A1) exhibited the highest affinity. Although JD-4 and JD-5 had high total read counts in the final round of sequencing, they showed poor abundance among the top 100 aptamer sequencing results as single-sequence family members, and their affinity performance was not outstanding.

During the screening process, OA was used as a counter-screening target to enhance specificity to some extent. To verify the specificity of the aptamers against multiple marine toxins, a wider variety of toxins were employed in the affinity detection, including the small molecule Saxitoxin (STX); OA and Dinophysistoxin (DTX) with similar molecular weights to azaspiracids; the slightly larger molecular weight Palytoxin (PLTX); and azaspiracid-1, which shares a similar backbone structure with azaspiracids but differs by only one methyl group (Figure 5). Preliminary affinity results indicated that the response values of the two high-affinity aptamers and the most abundant aptamer JD-1 to OA, DTX, STX, and PLTX were all below 0.1 nm. This suggests that the initially truncated aptamers have a certain effect in distinguishing between OA, DTX, STX, and PLTX, but their ability to differentiate between AZA-1 and AZA-2 is relatively poor (Figure 6).

### 2.2. Docking and Truncation Optimization of Aptamers

According to the results of autodock molecular docking, we can obtain binding sites with the lowest docking free energy to azaspiracids-2 for each aptamer. By removing sequences unrelated to the docking sites, including truncating the 5′-end of the aptamer (abbreviated as RM5 hereafter), removing the 3′-end of the aptamer (RM3), deleting internal sequences (RMM), and performing base substitutions based on the screening results from high-throughput sequencing, we optimized and verified the binding affinities of dozens of aptamers. We finally obtained two high affinity aptamers, JD2-RM3-27C28T (Figure 7) with *K*_D_ = 8.7 × 10^−8^ M and JD3-RMM1 with *K*_D_ = 6.8 × 10^−8^ M (Figure 8) (Table A1).

### 2.3. Molecular Dynamics Simulation of Candidate Aptamers

Subsequently, conventional dynamic simulations under an all-atom water model for 100 ns were conducted on the modified high-affinity aptamers JD2-RM3-27C28T and JD3-RMM1. The relative positional relationships between the aptamers and toxins over the 100 ns period are presented in Figure 9 and Figure 10.

The commonly used method for evaluating the stability of molecular structures in dynamic simulations is to calculate the root mean square deviation (RMSD) of the initial simulated structure (Figure 11). Although the JD-2 aptamer is in equilibrium, its RMSD value showed steep changes during some simulation time periods, indicating that the toxin ligand may have undergone unstable state changes during the simulation process. After simulation, JD2-RM3-27C28T and JD3-RMM1 reached equilibrium within a few nanoseconds and fluctuated slightly within the range of equilibrium values. Combined with the relative positions of toxin aptamers in the simulation process shown in Figure 9 and Figure 10, the stability of the simulation state was further demonstrated. The multiple systems validated during the simulation process achieved stability within the specified simulation duration, while also verifying the rationality of the applied force field and parameters.

Figure 12 shows the root mean square fluctuation (RMSF) during the simulation of different aptamers and toxins. The RMSF value measures the amplitude of all atoms deviating from the average position, reflecting the flexibility and activity intensity of each nucleotide of the aptamer in the simulation. Among them, the 3′ end of the JD-2 aptamer showed a significant (2 nm) shift in some nucleotides, and other atoms also had a variation amplitude of about 1 nm (Figure 12A). However, in the aptamer JD2-RM3-27C28T, only a few free adenosine nucleotides at the 5′ end and the corresponding number of atoms at the middle nucleotides T12, G13, T14, and T15 shifted by 2 Å–6 Å, while the RMSF values of the nucleic acids at G5, A6, T17, and C18 that interact with the aptamer were all below 2 Å, indicating improved stability of the aptamer nucleotides (Figure 12B). In JD3-RMM1, the nucleotide region that binds to the target is relatively stable, while the nucleotide corresponding to the intermediate circular structure formed by non-docking exhibits significant fluctuations (Figure 12C). The binding of the aptamer to the target promotes the stability of the local region.

The radius of gyration (Rg) of the three further illustrates the simulated state. The Rg of the unmodified aptamer JD-2 fluctuated between 1 nm and 3 nm (Figure 13A), while JD2-RM3-27C28T remained stable between 1.9 nm and 2.0 nm after 25 ns (Figure 13B). The simulation animation revealed that at the beginning of docking, the HIJK ring of the AZA-2 was in a state far from the aptamer (Figure 9), while in the later stage, the first two ends of the AZA-2 were further approached, resulting in a decrease in the cyclotron radius at 25 ns. JD3-RMM1 remained stable between 1.2 nm and 1.5 nm (Figure 13C).

As the simulation progresses, the solvent accessible surface area of the aptamer gradually decreases (Figure 14), indicating that the binding sites exposed by the binding of the target to the aptamer gradually decrease, and the binding stability of the aptamer to the target gradually improves.

The use of graphical representation to depict changes in molecular conformational free energy can generate a free energy landscape diagram (Figure 15), in which the horizontal axis (PC1) and the vertical axis (PC2), respectively, represent RMSD (root mean square deviation) and Rg (radius of gyration) from the primary simulation results. The different dark regions in the three simulations represent distinct stable conformations. The dark region for the JD-2 aptamer is significantly larger than that for the modified aptamers (Figure 15A), and the continuous reduction in area indicates a gradual increase in aptamer stability and a decrease in conformational change values. The concentration of dark regions in the two modified aptamers, JD2-RM3-27C28T (Figure 15B) and JD3-RMM1 (Figure 15C), suggests the presence of only one stable conformation. However, the deep stable state of the unmodified JD-2 aptamer has multiple positions with similar distances, indicating the possible existence of multiple approximate conformations within the stable structure, with relatively lower difficulty in structural changes. The two modified aptamers still exhibit excellent specificity without any loss thereof (Figure 16).

### 2.4. Basic Testing of Sensor Performance

After substituting the detection data from the previously validated aptamers into the sigmoidal logistic four-parameter equation, the JD2-RM3-27C28 curve and the JD3-RMM1 curve were fitted separately for the two aptamers (Figure 17). The linear regression equations for JD2-RM3-27C28 is shown in Figure 18A, its linear detection range is between 100 and ~1000 nM. For JD3-RMM1, shown in Figure 18B, its linear detection range is between 100 and 1000 nM (Figure 18). After calculation, the LOD and LOQ of the JD2-RM3-27C28 sensor are 39 nM and 131 nM, respectively. The LOD and LOQ of the JD3-RMM1 sensor are 34 nM and 116 nM, respectively.

Regarding the reusability of the sensor, it decays by about 21.2% after the first 20 cycles of use. As the number of reactions gradually increases, the degree of decay gradually decreases, and the degree of decay after the last 80 cycles is about 11.25%, indicating that this sensor can achieve stable multiple sample detections within a hundred cycles. The relative standard deviation (RSD) is 4.77%, indicating a low degree of data dispersion and good consistency (Figure 19). Moreover, for monitoring multiple samples, the detection interval only needs 60 s to be used for the next sample.

Regarding the specificity of these two aptamers in detecting non-azaspiracid toxins, they exhibit good specificity towards STX, DTX, OA, and PLTX. However, there is some cross-reactivity concerning distinguishing AZA-2 from similar toxins that share the same backbone but differ in the number of methyl groups (Figure 20).

The actual sample detection results are shown in Table 2, with a stable recovery rate between 92% and 101% and RSD ranging from 1.80% to 5.54%. The accuracy and precision meet the standard, which proves that impurities in water samples have no significant impact on the detection results in our laboratory environment.

Therefore, both sensors have an appropriate linear range, high specificity, good repeatability, and low LOD and LOQ values, which can be applied to the preliminary detection of AZA-2 in environmental samples.

For the three environmental samples, the response values showed low or no change compared to the control response values, exceeding the minimum value of the linear detection line, indicating that the sensor did not detect any toxic components in the three hydrological samples (Table 3).

## 3. Discussion

The choice of the Capture-SELEX method is justified as it avoids direct chemical modification of the azaspiracid (AZA-2), thereby preserving the toxin’s natural structure and activity. This is crucial for ensuring that the selected aptamers possess high specificity and affinity.

This experiment experiences library loss at multiple steps. Factors such as the richness of library synthesis, the immobilization rate of the library not reaching 100%, base bias in PCR, inefficient recovery methods, qPCR prior to high-throughput sequencing, and decomposition during library collection and sample submission can all reduce library richness. These steps offer significant potential and importance for enhancing library richness during the screening process. At the same time, applying reasonable screening pressure to reduce non-specific binding is also necessary. During the screening process, methods such as reducing the number of magnetic beads, lowering library concentration, and shortening incubation time are used to increase screening pressure, which helps improve the specificity of aptamers [48]. The choice of okadaic acid (OA) as a counter-selection target is reasonable because it belongs to the same class of polyether marine toxins as AZA-2 and shares similar chemical structures and molecular weights. The impact of counter-selection target choice on aptamer specificity is discussed below.

SELEX identifies the top 10 enriched sequences from the final round of sequence analysis. In the initial rounds, some sequences already rank high in read counts. However, this does not mean that we can directly perform one-round SELEX, as there are still many complex structures that require further competitive elimination. Moreover, due to the similar structures generated by similar sequences, there are subtle differences in affinity and specificity that still require additional rounds of elimination to distinguish.

Although the selected aptamers exhibit high affinity and specificity, they may contain some non-essential nucleotides. These nucleotides not only increase the length and complexity of the aptamers but may also reduce their binding efficiency to the target [49]. Therefore, optimizing the aptamers is necessary. In this study, we truncated the aptamers by shortening the primers at both ends and further validated the effectiveness of the optimization strategy through molecular docking and dynamics simulations. The aptamer optimization in this study primarily focused on truncation. During truncation, as many existing sequences as possible were removed without disrupting existing base pairings. If changes in the docking site or a significant decrease in affinity occurred, we referred to the sequences in the selected library for backfilling or designed by adding or reducing paired base sequences. This process did not involve large-scale affinity measurements of aptamers, which may have resulted in the loss of high-affinity aptamers. Additionally, the optimization process did not employ other methods, which also reduced the likelihood of obtaining aptamers with higher affinity. Methods such as phosphate backbone modification, sugar ring modification, optimization of the binding environment, and combining aptamers with other functional molecules (such as antibody fragments, nanomaterials) can further enhance the affinity of aptamers for their targets by improving nuclease resistance and thermal stability [50].

Docking results can only provide a reference for truncation optimization. Sometimes, using several random aptamer sequences for docking can yield docking results, but these may not have practical research applications. Ultimately, it is necessary to revert to affinity detection methods such as biolayer interferometry (BLI), microscale thermophoresis (MST) [51], surface plasmon resonance (SPR) [52], isothermal titration calorimetry (ITC) [53], and fluorescence resonance energy transfer (FRET) [54] for validation. Different affinity characterization methods are not universally applicable. This experiment only used a single molecular detection method, and incorporating more affinity characterization methods could further enhance credibility.

Although the study preliminarily explored the binding mechanism between aptamers and AZA-2 using computer simulations, the research exploring the mechanism is not yet in-depth. In the future, multiple technical means, such as X-ray crystallography and nuclear magnetic resonance, should be further employed to deeply analyze the interaction mechanism between aptamers and their targets [55,56].

The application of spike detection in this experiment is highly persuasive. The sensitivity of the sensor designed in this experiment is comparable to that of ELISA, but its sensitivity cannot match that of the LC-MS detection method (Table 4). However, the detection results in environmental samples may be less convincing. 1. The distribution of toxins, especially marine biotoxins, in the environment is inherently sparse, and the distribution and occurrence of azaspiracids in China are relatively rare, reducing the likelihood of toxin detection. 2. The lack of positive control detection results based on existing detection methods such as LC-MS further reduces persuasiveness. In the future, it is necessary to learn relevant technologies and achieve controlled detection. 3. Sampling during algal blooms or harmful algal outbreaks may help improve the detection significance and expected detection rate. Additionally, the sensor developed in this experiment has not been tested in replicate environmental samples. In the future, the detection performance of the sensor under complex environments can be evaluated to further improve the affinity and specificity of aptamers, optimize the detection sensitivity and stability of the sensor, and expand its application scope.

## 4. Materials and Methods

### 4.1. Chemicals and Reagents

AZA-2 was purchased from the National Research Council of Canada (Canada). OA, STX, DTX, and PLTX were purchased from Taiwan Algal Science Inc. (Taiwan, China). All ssDNA oligonucleotides were synthesized and purified by Sangon Biotechnology Co. Ltd. (Shanghai, China). GoTaqHot^®^ Start Colorless Master Mix was purchased from Promega Corporation (Madison, WI, USA). The QIAEX^®^ II Gel Extraction Kit was obtained from Qiagen (Frankfurt, Germany). SA sensors were purchased from ForteBio (Shanghai, China). The selection buffer (SB) (0.9 mM CaCl_2_, 2.7 mM KCl, 1.5 mM KH_2_PO_4_, 0.6 mM MgCl_2_·6H_2_O, 0.1 M NaCl, and 20 mM Na_2_HPO_4_; pH = 7.4) was purchased from Tiandz (Beijing, China). The SB was used for aptamer screening, the BLI assay, and the performance evaluation of the sensor.

### 4.2. Aptamers Selection In Vitro

This screening uses a fixed sequence of 20 nt at the beginning and end, a random sequence of 40 nt in the middle, and a library with a total length of 80 nt for screening. The fixed nucleotides are the primer binding sites F1 and R1 for amplifying the library (Table 5). In addition, the fixed areas at both ends can form complementary feature-pairing of 10 bp. The design of R1 with a polyadenine tail (A_20_ = 20 nt A) facilitates the separation of ssDNA libraries into two different sizes of PCR products The library can be combined with the capture sequence through slow refolding, and the capture sequence (Table 5 Cap1) is bound to magnetic beads through streptavidin to achieve the binding of the three. High-affinity aptamers can be obtained by incubating with positive and negative screening targets. After PCR amplification of the library and recovery of the electrophoresis gel, the next round of library is obtained. The collected library is amplified by qPCR and subjected to high-throughput sequencing to obtain the aptamer sequence.

### 4.3. Affinity Determination

The aptamer sequence is labeled with biotin at the 5′ end to enable it to be immobilized on the probe of a biofilm interferometer (BLI) for affinity determination. When using BLI, the first step is to load the program: ① Baseline (1 min)—choose a 96-well plate as the buffer solution for balancing baseline measurements, and select the shaking mode when using the probe to thoroughly clean it. ② Loading (3 min)—the 96-well plate to be selected contains a final concentration of 2 μmol/L of biotin modified 5′ end aptamer that has been diluted in buffer solution. At the same time, when using the probe, the shaking mode is selected to fully bind the aptamer and remove the loosely bound aptamer. ③ Baseline (1 min)—the 96-well plate to be selected is the buffer solution to remove unbound aptamers on the sensor surface and stabilize the baseline. Next, combine the program: ① Baseline (1 min)—the 96-well plate to be selected is used as a buffer to remove unbound aptamers on the sensor surface and stabilize the baseline. ② Association (3 min)—the 96-well plate to be selected is a key step for the binding of toxins and aptamers diluted in buffer solution. ③ Dissociation (3 min)—the 96-well plate to be selected is the buffer solution for the dissociation stage of the aptamer.

### 4.4. Molecular Docking

Using RNAfold web server “http://rna.tbi.univie.ac.at/cgi-bin/RNAWebSuite/RNAfold.cgi (accessed on 3 May 2024)” for predicting the secondary structure of single-stranded nucleic acids, the sequence is first inputted in the order of 5′ to 3′. The prediction conditions are a folding temperature of 25 °C and ion strength equal to the concentration in the screening buffer; the generation of the tertiary structure depends on the RNAfold website for predicting the secondary structure of nucleic acid aptamers. During prediction, the DNA sequence needs to be modified to the RNA sequence. When predicting, choose the minimum free energy and allocation function while avoiding the formation of isolated base pairs. The generated dot bracket form and RNA sequence input into RNA Composer can obtain the tertiary structure of the sequence, and the file is saved in pdb format. Entering it into Discovery Studio (Ver4.5) for mutation can convert RNA into DNA. For small molecules, ChemSpider can be used “https://www.chemspider.com/ (accessed on 12 Decenmber 2023)”: search for the CAS number to obtain the mol format file, obtain the aptamer and toxin pdb files, and use Autodock 1.5.7 software for molecular docking. Set the center of the small molecule in the software and detect the twist key, and then set up the docking box. After setting up the docking box, use Dejavu GUI to move the small molecule outside the box and perform grid. Before formal docking, set the docking parameters, use parameter genetic algorithm, and define the nucleic acid as a rigid structure. Ten minimum energy docking methods are used for each nucleic acid structure and small molecule, and the number of hydrogen bonds formed are counted.

### 4.5. Molecular Dynamic Simulation

#### 4.5.1. Obtaining Simulated Files

Using the GROMACS 2018.8 software based on the LINUX system for molecular dynamics simulation, the top file, gro file, and itp file are first generated in the amber14sb_oL15. ff force field file. Due to software differences, there may be inconsistencies between the courtyard naming and the rtp atom naming in the force field file when using gmx, which requires manual modification in the pdb file. Use Mg^2+^ and Cl^-^ ions for cation–anion balance in charge balancing. The processing method for ligands is to use acpype to generate atomic topology under GAFF force field for 134 ligand atoms in the top file. Before using acpype, an AmberTools23 environment should be established and small molecules should be hydrogenated to calculate all atomic charges. Extract the IPT, top, and gro files with GMX suffixes and run them in the Gromacs software. Merge them with nucleic acid molecules and perform box setting, solvent addition, equilibrium charge, and other operations. Next, use MDP files for energy minimization, restrictive dynamics, and conventional dynamics. When running energy level minimization, sometimes cg (conjugate gradient method) mode is used to process complex structures that cannot be processed within a limited number of steps. To achieve equilibrium (<100 KJ/mol/nm), energy reduction can be achieved by combining the steepest descent method with cg mode.

#### 4.5.2. Output of Dynamic Results

The root mean square deviation (RMSD) value obtained using the ”gmx rms” command can reflect the stability of the system during the simulation process. Using ”gmx rmsf” to obtain root mean square fluctuation (RMSF) values, RMSF calculates the fluctuation of each atom relative to its average position, characterizing the average value of structural changes over time. Using ”gmx gyrate” to analyze the changes in the protein’s radius of rotation, the smaller the radius, the better the density. Using ”gmx sasa” to analyze the solvate accessible surface area (SASA) of solvents is an important means of describing the hydrophobicity of nucleic acids, and the hydrophobicity of nucleic acid residues may be an important physical effect affecting the conformational folding of nucleic acids. Use DulvyTools to draw free energy morphology maps and display these using ”gmx sham”.

### 4.6. Preliminary Practice of Sensors

#### 4.6.1. Basic Testing of Sensor Performance

Using GraphPad Prism 9.5 software, we fit the data of the original concentration of polymyxin toxin and BLI response values to a four-parameter sigmoidal logistic nonlinear regression. In the equation, R_max_ and R_min_ represent the maximum and minimum response values, respectively. EC_50_ is the AZA-2 concentration corresponding to half of the maximum response value, and b is the Hill slope of the curve:Y = (R_max_ − R_min_)/[(1 + (X/EC_50_)^b^)] + R_min_(1)

The limit of detection (LOD) refers to the lowest concentration or amount of a substance that can be identified and confirmed to be present in a sample under specific experimental conditions with a given confidence level. It does not require quantitative measurement, as long as it is indicated to be above or below the specified concentration. Calculate the response values of 20 test buffer solutions using an aptamer sensor, and obtain the LOD of the sensor according to the formula LOD = 3 Sa/b. The limit of quantification (LOQ) refers to the minimum amount of the analyte in a sample that can be quantitatively determined, usually determined by the signal-to-noise ratio method. Calculate the LOQ of the sensor based on the formula and LOQ = 10 Sa/b. Among them, Sa represents the standard deviation of the response values of 20 blank samples, and b represents the slope of the calibration curve. Under the same experimental conditions, AZA samples were repeatedly tested 100 times using the same aptamer sensor, and the response values were recorded and the RSD values were calculated. Specific detection was performed using five toxins, AZA-1, OA, DTX, PLTX, and STX, at the same concentration as the AZA-2 standard.

#### 4.6.2. Preliminary Testing of Spiked Samples and Hydrological Samples

Firstly, take 1 mL of seawater and tap water each, remove bacteria and impurities through a 0.45 μm filter, add a bottle of toxin standard to it, evaporate at room temperature, and then add 100 μL of buffer solution to resuspend, obtaining a concentrated seawater or tap water sample containing toxins with a concentration of 7.15 μM. Store it at 4 °C for future use. To further validate the toxin detection capability of the sensor, three hydrological samples were selected for verification.

### 4.7. Statistical Analysis

Statistical analysis was performed using Microsoft Excel 365 or GraphPad Prism 9.5. The data were subjected to analysis by Student’s *t*-test using SPSS version 22.0. The mean differences were considered significant at *p* < 0.05.

## 5. Conclusions

This study focuses on the screening and identification of ssDNA aptamers for the marine polyether small-molecule biotoxin AZA-2. JD2-RM3-27C28T with a dissociation constant (*K*_D_) of 8.7 × 10^−8^ nM and JD3-RMM1 with a *K*_D_ of 6.8 × 10^−8^ nM were obtained, showing a two order-of-magnitude improvement in affinity compared to the pre-optimized aptamers. The optimized aptamers exhibited significant enhancements in both affinity and specificity, and an aptamer sensor based on BLI technology was successfully constructed. These sensors demonstrate high sensitivity, specificity, and reproducibility, enabling rapid and repeatable detection of AZA-2 in environmental samples, and its detection sensitivity surpasses that of existing enzyme-linked immunosorbent assays (ELISAs) and mouse bioassays.

In summary, this study successfully developed a BLI-based AZA-2 aptamer sensor, providing a new technical approach for the rapid detection of marine biotoxins. This innovative technology not only improves detection efficiency and accuracy but also reduces detection costs, offering strong technical support for marine food safety, environmental protection, and public health protection. The combined use of these techniques and methods not only enhances screening efficiency and accuracy but also provides robust support for aptamer optimization and validation. This study offers valuable references and insights for subsequent nucleic acid and protein research endeavors.

## Figures and Tables

**Figure 1 marinedrugs-23-00183-f001:**
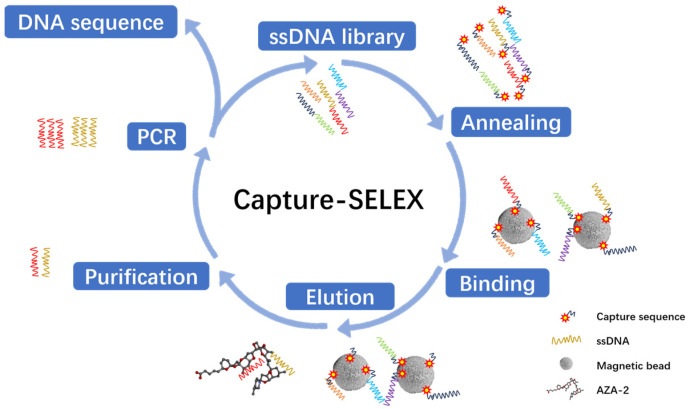
Schematic diagram of the Capture-SELEX screening aptamer process.

**Figure 2 marinedrugs-23-00183-f002:**
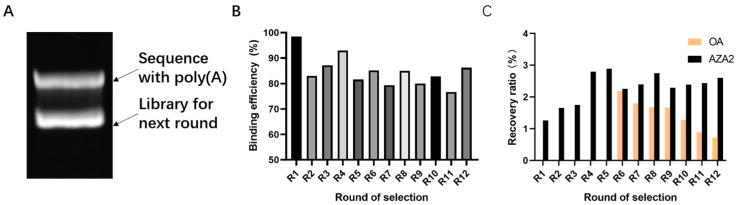
Monitoring data during the Capture-SELEX process. (**A**) The binding efficiency of each selection round. (**B**) Recovery ratio of ssDNA for each selection round of targets. (**C**) PCR results from the long short-chain separation method.

**Figure 3 marinedrugs-23-00183-f003:**
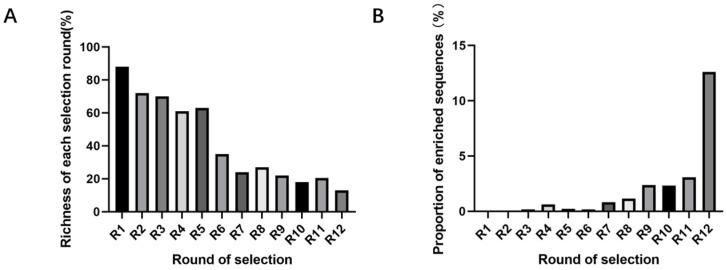
Richness of each selection round and proportion of enriched sequences. (**A**) Richness of each selection round. (**B**) Proportion of enriched sequences of the top 10 sequences in the pool of different rounds.

**Figure 4 marinedrugs-23-00183-f004:**
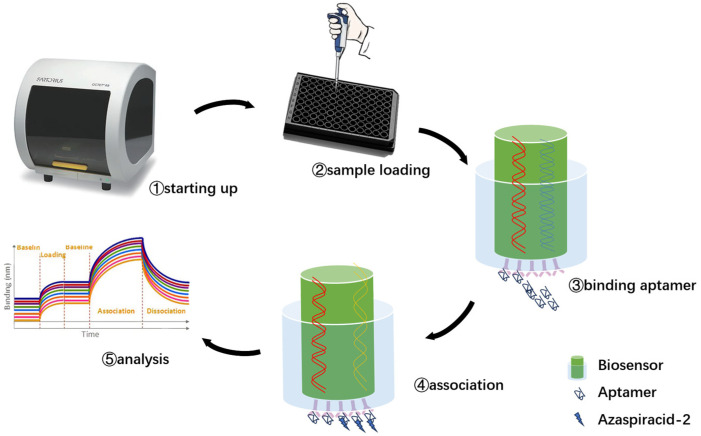
Basic operating procedures of BLI.

**Figure 5 marinedrugs-23-00183-f005:**
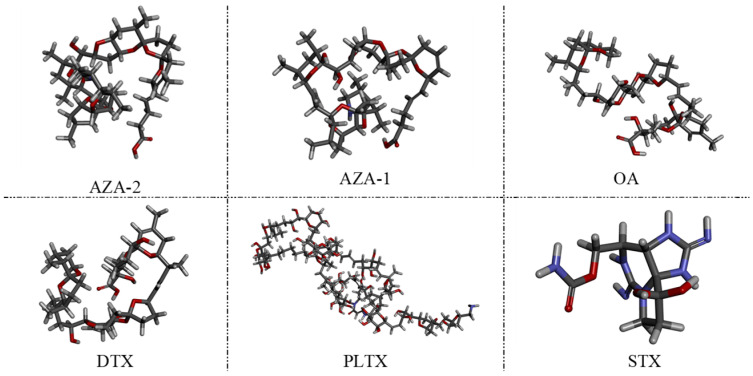
Three-dimensional structure of AZA-2, AZA-1, OA, DTX, PLTX, and STX.

**Figure 6 marinedrugs-23-00183-f006:**
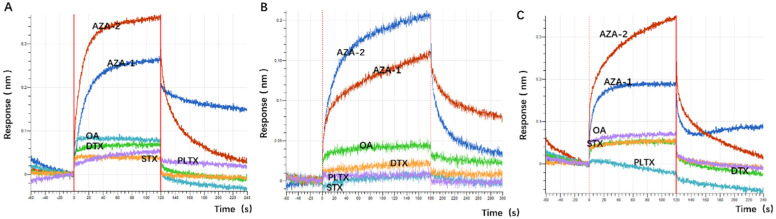
The specificity of some family aptamers. (**A**) BLI specificity results for aptamer JD-1. (**B**) BLI specificity results for aptamer JD-2. (**C**) BLI specificity results for aptamer JD-3.

**Figure 7 marinedrugs-23-00183-f007:**
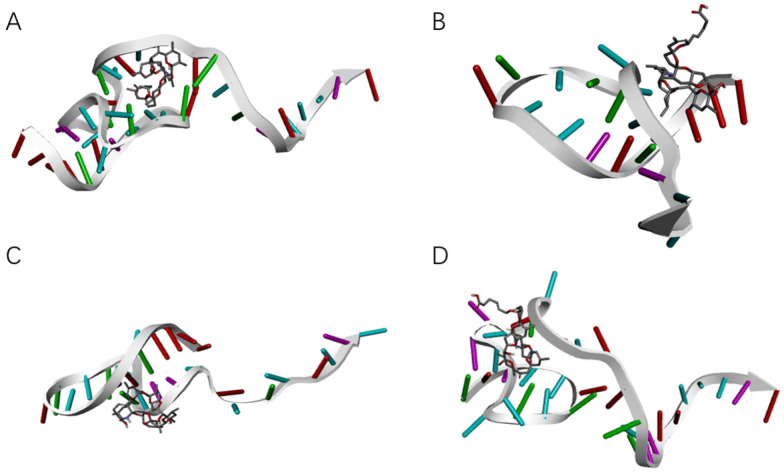
Docking results for aptamer JD-2 and its modifications. (**A**) Docking site between aptamer JD-2 and AZA-2. (**B**) Docking site between aptamer JD2-RM3 and AZA-2. (**C**) Docking site between aptamer JD2-RM3-27C28T and AZA-2. (**D**) Docking site between aptamer JD2-RM5 and AZA-2.

**Figure 8 marinedrugs-23-00183-f008:**
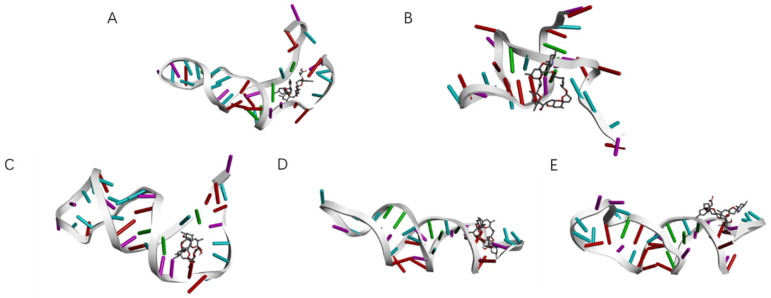
Docking sites of JD-3 family aptamers. (**A**) Docking site between aptamer JD-3 and AZA-2. (**B**) Docking site between aptamer JD3-RMM1 and AZA-2. (**C**) Docking site between aptamer JD-8 and AZA-2. (**D**) Docking site between aptamer JD-9 and AZA-2. (**E**) Docking site between aptamer JD-10 and AZA-2.

**Figure 9 marinedrugs-23-00183-f009:**
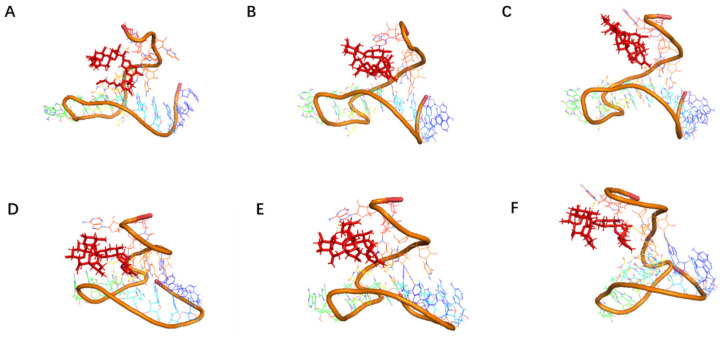
The positional relationship between small molecule toxins and aptamer JD2-RM3-27C28T displayed at different simulation times. (**A**) The relative position between the aptamer JD2-RM3-27C28T and the AZA-2 at 10 ns. (**B**) The relative position between the aptamer JD2-RM3-27C28T and the AZA-2 at 20 ns. (**C**) The relative position between the aptamer JD2-RM3-27C28T and the AZA-2 at 50 ns. (**D**) The relative position between the aptamer JD2-RM3-27C28T and the AZA-2 at 60 ns. (**E**) The relative position between the aptamer JD2-RM3-27C28T and the AZA-2 at 70 ns. (**F**) The relative position between the aptamer JD2-RM3-27C28T and the AZA-2 at 100 ns.

**Figure 10 marinedrugs-23-00183-f010:**
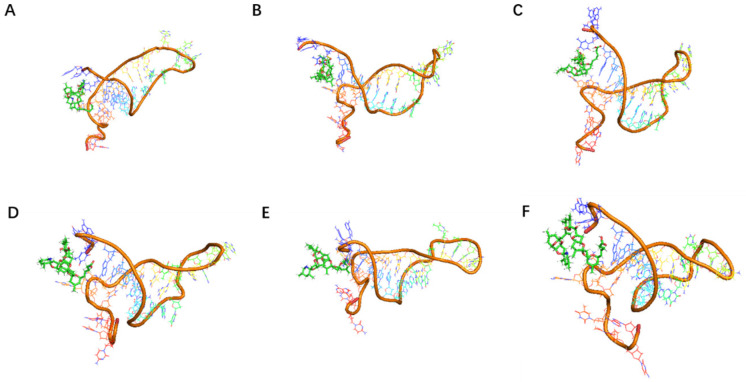
The positional relationship between small molecule toxins and aptamer JD3-RMM1 displayed at different simulation times. (**A**) The relative position between the aptamer JD3-RMM1 and the AZA-2 at 10 ns. (**B**) The relative position between the aptamer JD3-RMM1 and the AZA-2 at 20 ns. (**C**) The relative position between the aptamer JD3-RMM1 and the AZA-2 at 50 ns. (**D**) The relative position between the aptamer JD3-RMM1 and the AZA-2 at 60 ns. (**E**) The relative position between the aptamer JD3-RMM1 and the AZA-2 at 70 ns. (**F**) The relative position between the aptamer JD3-RMM1 and the AZA-2 at 100 ns.

**Figure 11 marinedrugs-23-00183-f011:**
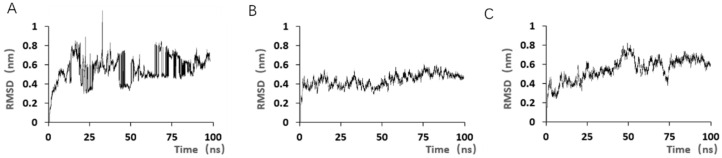
RMSD results for the 100 ns simulation with different modified aptamers. (**A**) RMSD of aptamer JD-2. (**B**) RMSD of aptamer JD2-RM3-27C28T. (**C**) RMSD of aptamer JD3-RMM1.

**Figure 12 marinedrugs-23-00183-f012:**
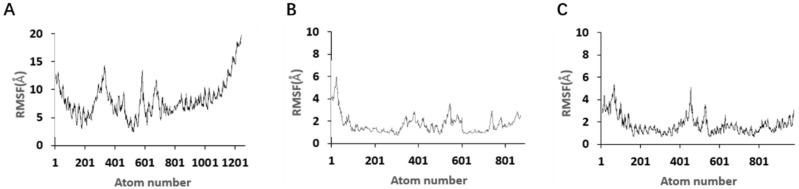
RMSF results for the 100 ns simulation with different modified aptamers. (**A**) RMSF of aptamer JD-2. (**B**) RMSF of aptamer JD2-RM3-27C28T. (**C**) RMSF of aptamer JD3-RMM1.

**Figure 13 marinedrugs-23-00183-f013:**
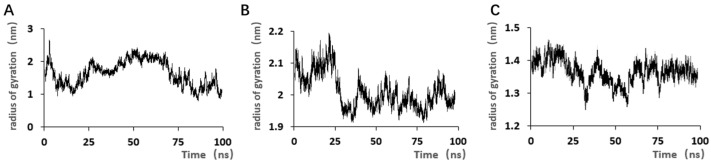
Radius of gyration results for the 100 ns simulation with different modified aptamers. (**A**) radius of gyration of aptamer JD-2. (**B**) radius of gyration of aptamer JD2-RM3-27C28T. (**C**) radius of gyration of aptamer JD3-RMM1.

**Figure 14 marinedrugs-23-00183-f014:**
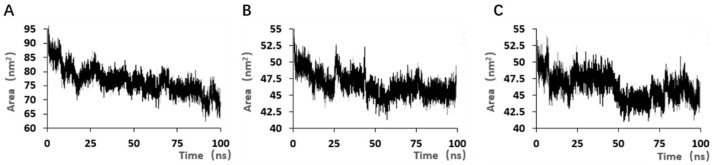
Solvate accessible surface area results for the 100 ns simulation with different modified aptamers. (**A**) Solvate accessible surface area of aptamer JD-2. (**B**) Solvate accessible surface area of aptamer JD2-RM3-27C28T. (**C**) Solvate accessible surface area of aptamer JD3-RMM1.

**Figure 15 marinedrugs-23-00183-f015:**
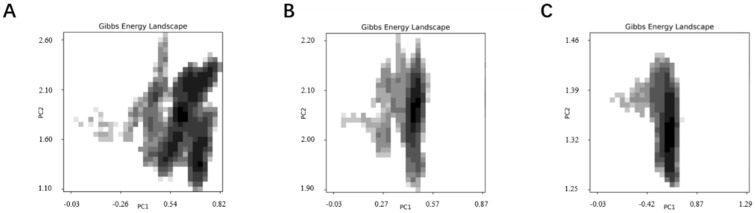
The 100 ns simulated free energy morphology of different modified aptamers. (**A**) Solvate accessible surface area of aptamer JD-2. (**B**) Solvate accessible surface area of aptamer JD2-RM3-27C28T. (**C**) Solvate accessible surface area of aptamer JD3-RMM1.

**Figure 16 marinedrugs-23-00183-f016:**
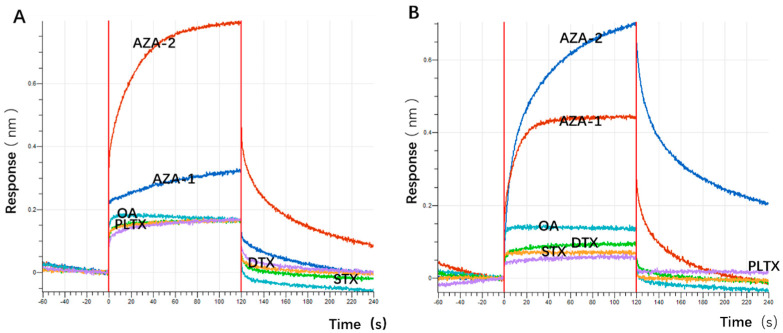
The specificity of some family aptamers. (**A**) BLI specificity results for aptamer JD-2-RM3-27C28T. (**B**) BLI specificity results for aptamer JD3-RMM1.

**Figure 17 marinedrugs-23-00183-f017:**
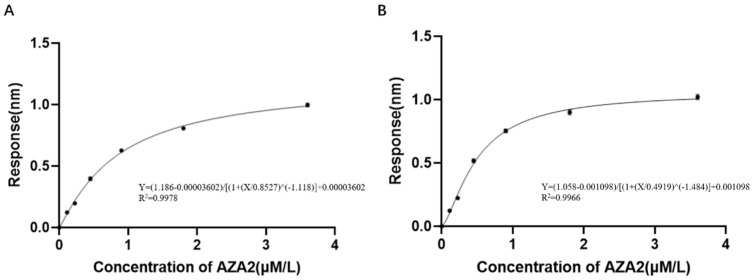
Fitting curves and equations of sigmoidal logistic nonlinear regression equations with different aptamers. (**A**) Fitting curve and equation of the sigmoidal logistic nonlinear regression equation for aptamer JD2-RM3-27C28T. (**B**) Fitting curve and equation of the sigmoidal logistic nonlinear regression equation for aptamer JD3-RMM1.

**Figure 18 marinedrugs-23-00183-f018:**
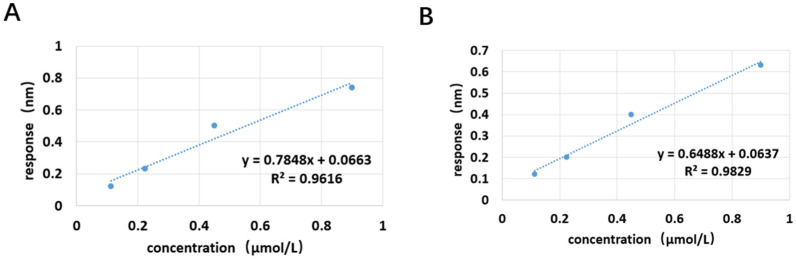
Linear interval fitting curve of aptamer sensor. (**A**) Linear interval fitting curve of the aptamer JD2-RM3-27C28T sensor. (**B**) Linear interval fitting curve of the aptamer JD3-RMM1 sensor.

**Figure 19 marinedrugs-23-00183-f019:**
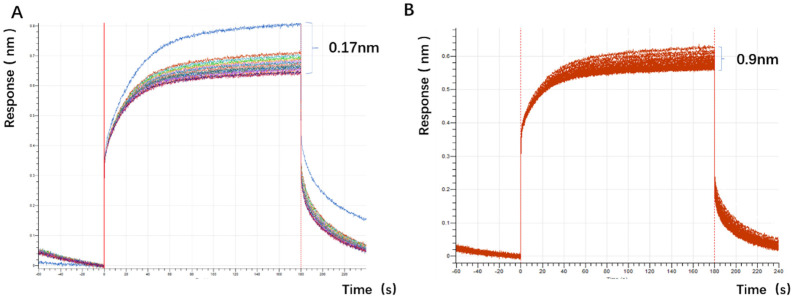
The degree of attenuation of sensors after multiple uses. (**A**) The degree of attenuation of the sensor in the first 20 cycles. (**B**) The degree of attenuation of the sensor between 21 and 100 times.

**Figure 20 marinedrugs-23-00183-f020:**
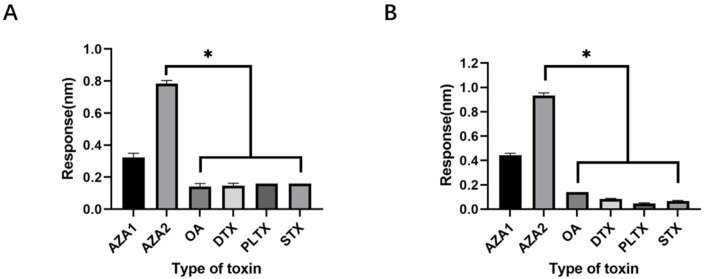
Specificity determination of different sensors. (**A**) Specificity of response values of aptamer JD2-RM3-27C28T to multiple toxins. (**B**) Specificity of response values of aptamer JD3-RMM1 to multiple toxins. * means *p* < 0.05 vs.AZA-2 (control).

**Table 1 marinedrugs-23-00183-t001:** AZA-2 aptamer sequences obtained by removing the primers at both ends.

Name	Sequence (5′-3′)
JD-1	GAATGGACCCGGTATAATTCCCTCAAGAGTGCCAATTTCA
JD-2	AAAAGACTTAGTGTTGTCCTATGTATAAGTGCCAATTTCA
JD-3	ACTAGTGCAAATCTATTCCTATGTTACAGTGCCAATTTCA
JD-4	GACATCGAGAAGAATCCTGATACGACTTGGCTTTGCTGGC
JD-5	CCAACATGATGTTCCGTCATTTTGAGGTGTGTACACCGTG
JD-6	GATGACACTTGTTTATGCCTATGATGATGTGCCAATTTCA
JD-7	GAATGGACCCGGTATAATTCCCTCGAGAGTGCCAATTTCA
JD-8	ACTAGTGCAAATCTATTCCTATGTTTCAGTGCCAATTTCA
JD-9	ACTAGTGCAAATCTATTCCTATGGTTCAGTGCCAATTTCA
JD-10	ACTAGTGCAAATCTATTCCTATGATACAGTGCCAATTTCA

**Table 2 marinedrugs-23-00183-t002:** Concentration detection of AZA-2 in actual spiked samples.

Sample	Spiked AZA-2 (μM)	Recovery Rate (%)	RSD (%)
Tap water	0.72	100.53	3.94
	1.43	99.39	1.80
Seawater	0.72	98.11	5.44
	1.43	92.3	2.00

**Table 3 marinedrugs-23-00183-t003:** Concentration detection of AZA-2 in hydrological samples.

Sample	Response (nm)	RSD (%)
Sea surface sample located 100 nautical miles east of Changxing Island in Shanghai	0.003	19.23
Seawater at the beach of Nanhui New City, Shanghai	0.001	0
Qiujiang Branch of the Huangpu River in Shanghai	0.000	0

**Table 4 marinedrugs-23-00183-t004:** Existing detection methods for azaspiracids and their LOD and LOQ values.

Year	Method	LOD	LOQ
2015	SPE- HPLC-MS [57]	0.013–0.085 μg/kg	1.00 μg/kg
2015	UHPLC-HR-Orbitrap-MS [58]	0.006–0.050 ng/mL	0.018–0.227 ng/mL
2017	MB-based direct immunoassay [59]	63 μg/kg	120–2875 μg/kg
2019	ELISA [30]	0.30–4.1 ng/mL	37 μg/kg
2019	MSPE-UPLC-MS [60]	0.4–1.0 μg/kg	1.0–4.0 μg/kg
2020	SPATT-UPLC-ESI-MS [61]	0.001–0.05 μg/L	0.04 μg/mL
2020	LC-MS [62]		0.3–0.4 μg/kg

**Table 5 marinedrugs-23-00183-t005:** List of all single stranded nucleotide sequences used.

Name	Sequence (5′-3′)
Lib	ATTGGCACTCCACGCATAGG-N40-CCTATGCGTGCTACCGTGAA [63]
F1	ATTGGCACTCCACGCATAGG
R1	A_20_-Spacer18-TTCACGGTAGCACGCATAGG
Cap1	CCTATGCGTGGAGTGCCAAT-biotin

## Data Availability

The data presented in this study are available on request from the corresponding author.

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
