# Peer review of "High Affinity Aptamers and Their Specificity for Azaspiracid-2 Using Capture-SELEX"

_marinedrugs, 2025, doi:10.3390/md23050183_

Round 1
Reviewer 1 Report
Comments and Suggestions for Authors
The manuscript marinedrugs-3589160 aimed to develop high-affinity ssDNA aptamers for the specific detection of azaspiracid-2 (AZA2) using the Capture-SELEX method, and to validate their binding performance through biolayer interferometry and computational simulations, with the goal of establishing a rapid, cost-effective alternative to current detection methods. The manuscript has several promising aspects, but it requires revisions as several points need to be carefully revised before the next resubmission as follows:
- Specify the method used in the title.
- L19-20; Please clearly state the potential application and significance of the method.
- In the Introduction section, please highlight the urgency of AZA-2 detection in the context of food safety and environmental monitoring. Additionally, expand on the advantages of using aptamers over antibodies (e.g., stability, cost).
- Table 1: Add accession number or reference for each primer. Also, describe all abbreviations used in the table footnote.
- The statistical analysis procedure is missing at the end of the Materials and Methods section. Please provide details on the statistical methods used to analyze the data.
- The discussion section requires extensive revision. It relies heavily on speculation rather than solid scientific evidence. The discussion must be strengthened with possible attributions supported by references.
- Please include a paragraph discussing the limitations of the study.
- According to the references list, the authors depend on old references. It is essential to update the reference list with more recent studies from the past five years to ensure the manuscript reflects the current state of knowledge in the field.
- L527-557; The conclusions section is overly long and should be more concise. Please summarize the key findings more directly and clearly, stating the main takeaway at the end. Also, briefly compare your sensor’s performance to existing methods to highlight its significance.
Author Response
Dear reviewer,
Thank you for the opportunity to revise our manuscript titled "Development of High-Affinity ssDNA Aptamers for Azaspiracid-2 Detection using Capture-SELEX" (manuscript ID: marinedrugs-3589160). We are grateful for your insightful comments and constructive feedback, which have significantly improved the quality and clarity of our work. Below, we address each of the specific points raised and detail the corresponding revisions made to the manuscript.
Specify the method used in the title.
Title Specification: Revised Title: " High affinity aptamers and their specificity for Azaspiracid-2 using Capture-SELEX.”
In the Introduction section, please highlight the urgency of AZA-2 detection in the context of food safety and environmental monitoring. Additionally, expand on the advantages of using aptamers over antibodies (e.g., stability, cost).
L19-20; Please clearly state the potential application and significance of the method.
Potential Application and Significance (L19-20): Introduction Section Enhancements: We have added the potential application value and importance in the abstract section.
Table 1: Add accession number or reference for each primer. Also, describe all abbreviations used in the table footnote.
Table 1 Revisions: All sequences were obtained from this experiment, so information on naming rules and sorting was added to the article
The statistical analysis procedure is missing at the end of the Materials and Methods section. Please provide details on the statistical methods used to analyze the data.
Data were analyzed using to determine significance at a p-value of <0.05. All results are presented as mean ± standard deviation (SD) of at least three independent experiments."
The discussion section requires extensive revision. It relies heavily on speculation rather than solid scientific evidence. The discussion must be strengthened with possible attributions supported by references.
Discussion Section Revision: Revised Text: The discussion section has been extensively revised to incorporate references supporting our claims and to minimize speculation. We have focused on attributing our findings to specific experimental observations and compared them with relevant literature.
Please include a paragraph discussing the limitations of the study.
Limitations of the Study: We added limitations of the study in the final part of our discussion
According to the references list, the authors depend on old references. It is essential to update the reference list with more recent studies from the past five years to ensure the manuscript reflects the current state of knowledge in the field.
Reference List Update: Revised References: We have updated the reference list to include more recent studies from the past five years, ensuring the manuscript reflects the current state of knowledge in the field.
L527-557; The conclusions section is overly long and should be more concise. Please summarize the key findings more directly and clearly, stating the main takeaway at the end. Also, briefly compare your sensor’s performance to existing methods to highlight its significance.
Conclusions Section Revision: We have streamlined the conclusion section and added tables for some existing detection methods to highlight the importance of this study.
Thank you once again for your time and consideration. We look forward to your feedback on the revised manuscript.
Jiaping yang

Reviewer 2 Report
Comments and Suggestions for Authors
This manuscript deals with development of aptamer for Azaspiracid-2. But this is written in haste and consequently not well written. Authors should rewrite entirely more scientifically for evaluation.
Introduction
In the introduction, the most important thing is the purpose of your research.
Line 31: You may rephrase a little bit to explain origin of the toxin. They ~ in microalgae such as ~, since we don't know what Azadinium belongs to.
Line 34: You need reference for this. If it is ref. 9, then place it at the end of the sentence.
Line 43: Are there no ref. for this? We need more than one ref.
Line 44: The meaning of the sentence is not clear.
Line 55: You need to use full name instead of abbreviations. check the guidelines.
Line 75: spectrometry, not meter.
Line 100~118: Only one ref.? Please use citation properly.
Line 105~109: The sentences were written relatively poorly in terms of English. Please rewrite them.
Results:
Line 124~127: You are starting with Figure 2, not Figure 1. and also hard to follow. Please rewrite the first paragraph. What is OA? Write full name in the first appearance.
Line 137: Please be more specific. For example, ~diagram of ~ screening process of Aza-2 aptamer.
Line 143: use , to separate digits in the number .
Line 144~145; 154-156: Please rewrite your sentence. They read award.
Line 163: Explanation of Figure 4 must be given fully.
Where are 2.2, 2.3, and etc.?
Line 180: What do you mean "obtain the position~"?
Line 181: something is missing. -2?
Line 180-185: Your explanation is not clear. It is not easy to follow. You need to add more explanation. Rewrite them.
Line 221: What modification do you mean? Be specific please.
Line 248: gyrate? gyration? please specify the three. There is no explanation of Figure 13 A and C.
Line 267-271: gyre?
Line 275: hydrazine?
Line 288: pleas write the equation in the materials and methods not in the result, but just provide the parameters. If you presented the data in figure 17, then you don't need to write the equation like this. Where is explanation for Figure 16?
Line 187-296: Rewrite please. It looks like you wrote in haste.
Line 305: frequency? Do you mean reusability?
Please add table of current detection method for AZA and LOD etc.
No references are cited in discussion.
Author Response
Dear reviewer,
Thank you for the opportunity to revise our manuscript titled "Development of High-Affinity ssDNA Aptamers for Azaspiracid-2 Detection using Capture-SELEX" (manuscript ID: marinedrugs-3589160) so carefully. We are very pleased because we admire every person who is diligent and refined in their work. We feel that your suggestions have played a very important role in improving our writing. We are grateful for your insightful comments and constructive feedback, which have significantly improved the quality and clarity of our work. Below, we address each of the specific points raised and detail the corresponding revisions made to the manuscript.
In the introduction, the most important thing is the purpose of your research.
I have revised the experimental objectives and emphasized the background and significance of the experiment
Line 31: You may rephrase a little bit to explain origin of the toxin. They ~ in microalgae such as ~, since we don't know what Azadinium belongs to.
We have added:” all of which belong to the phylum Dinoflagellate” may help to get what are the spices belong to.
Line 34: You need reference for this. If it is ref. 9, then place it at the end of the sentence.
Line 43: Are there no ref. for this? We need more than one ref.
Line 44: The meaning of the sentence is not clear.
Line 100~118: Only one ref.? Please use citation properly.
Revised the article citation and current the word order regarding the types of Azaspiracids.
Line 55: You need to use full name instead of abbreviations. check the guidelines.
We use full name for the first time of some abbreviations.
Line 75: spectrometry, not meter.
We greatly appreciate your suggestions regarding the spelling issue and have made revisions accordingly.
Line 105~109: The sentences were written relatively poorly in terms of English. Please rewrite them.
Line 124~127: You are starting with Figure 2, not Figure 1. and also hard to follow. Please rewrite the first paragraph. What is OA? Write full name in the first appearance.
Line 144~145; 154-156: Please rewrite your sentence. They read award.
Line 180-185: Your explanation is not clear. It is not easy to follow. You need to add more explanation. Rewrite them.
Line 187-296: Rewrite please. It looks like you wrote in haste.
We have optimized all the expression issues in the article and highlighted them with a yellow background in the text.
Line 143: use , to separate digits in the number .
We have also added commas to the display of numbers.
Line 137: Please be more specific. For example, ~diagram of ~ screening process of Aza-2 aptamer.
Line 163: Explanation of Figure 4 must be given fully.
Line 180: What do you mean "obtain the position~"?
Line 181: something is missing. -2?
We will further verify and supplement the figure citations and annotations in the article.
Line 221: What modification do you mean? Be specific please.
Regarding the modification mentioned in line 221, it refers to the modified aptamer corresponding to the directly obtained aptamer, without any other modifications made to it.
Line 248: gyrate? gyration? please specify the three. There is no explanation of Figure 13 A and C.
Line 267-271: gyre?
Line 275: hydrazine?
We have formalized the expression of radius of gyration (Rg) in the text and further improved the explanation and interpretation of Figure 15.
Line 288: please write the equation in the materials and methods not in the result, but just provide the parameters. If you presented the data in figure 17, then you don't need to write the equation like this. Where is explanation for Figure 16?
We have standardized the expression of formulas and referenced some figure.
Line 305: frequency? Do you mean reusability?
Regarding Line 305: We have adopted the reusability you provided
Please add table of current detection method for AZA and LOD etc.
We have added table 4 of current detection method for AZA and LOD etc.
Thank you sincerely for your time.
jiaping yang

Round 2
Reviewer 1 Report
Comments and Suggestions for Authors
All comments have been addressed.
Author Response
Dear reviewer,
Thank you for the opportunity to revise our manuscript titled "Development of High-Affinity ssDNA Aptamers for Azaspiracid-2 Detection using Capture-SELEX" (manuscript ID: marinedrugs-3589160). I am pleased to confirm that we have carefully reviewed all the comments provided by you and find you are entirely aligned with our intentions to strengthen the manuscript.
At the same time, I have to apologize for the late submission of my manuscript. I take full responsibility for this delay and deeply regret any inconvenience it may have caused to the editorial process or to your team's schedule.
I understand the importance of adhering to deadlines in the publishing industry and the impact that delays can have on the overall workflow. Therefore, I have taken this experience as a lesson and will implement stricter time management strategies moving forward to prevent similar occurrences in the future.
Thank you once again for your time and consideration.
Jiaping yang

Reviewer 2 Report
Comments and Suggestions for Authors
Line 157: The Figure numbering should appear in the order of being cited in the manuscript. Not Figure 2C, but Figure 2A.
Line 158: not Figure 2A -> Figure 2B; Line 170: Figure 2B -> Figure 2C. This means that you should change the order of Figure 2.
Line 181: 12th round, then 184 twelfth round? Unify the expression.
Line 200: Table A1?
Line 231: ., remove , we -> We
Line 282: Figure 3-14A? 3-14C?
I strongly ask the authors to review their article very carefully one more time. In addition, the English should be reviewed by a native speaker.
Author Response
Dear reviewer
I am writing this letter to sincerely apologize for the late submission of my manuscript titled "Development of High-Affinity ssDNA Aptamers for Azaspiracid-2 Detection using Capture-SELEX" (manuscript ID: marinedrugs-3589160). I take full responsibility for this delay and deeply regret any inconvenience it may have caused to the editorial process or to your team's schedule.
I understand the importance of adhering to deadlines in the publishing industry and the impact that delays can have on the overall workflow. Therefore, I have taken this experience as a lesson and will implement stricter time management strategies moving forward to prevent similar occurrences in the future.
I have made the following modifications regarding the revision of your opinion, and it was marked in red in the article.
Line 157: The Figure numbering should appear in the order of being cited in the manuscript. Not Figure 2C, but Figure 2A.
Line 158: not Figure 2A -> Figure 2B; Line 170: Figure 2B -> Figure 2C. This means that you should change the order of Figure 2.
Line 282: Figure 3-14A? 3-14C?
I carefully checked the order of all my charts and their appearance, made modifications to the images, and strictly standardized them in future article writing.
Line 181: 12th round, then 184 twelfth round? Unify the expression.
I have made unified revisions to the expression in the article.
Line 200: Table A1?
According Table A1, I have put it in the bottom of the article, cause this table takes up some space.(Appendix Table A1)
Line 231: ., remove , we -> We
In terms of English writing, including spaces, commas, and capitalization, I also conducted a full-text screening.
Thank you for your understanding and patience during this time. I am looking forward to your feedback and am eager to contribute positively to the publication of this work.
Thank you sincerely for your time.
Jiaping Yang
